# Ultrasound Shear Wave Elastography, Shear Wave Dispersion and Attenuation Imaging of Pediatric Liver Disease with Histological Correlation

**DOI:** 10.3390/children9050692

**Published:** 2022-05-09

**Authors:** Ivan Cetinic, Charlotte de Lange, Yvonne Simrén, Nils Ekvall, Maja Östling, Liselotte Stén, Håkan Boström, Kerstin Lagerstrand, Hanna Hebelka

**Affiliations:** 1Department of Radiology & Pediatric Radiology, Sahlgrenska University Hospital, 41345 Gothenburg, Sweden; charlotte.de.lange@vgregion.se (C.d.L.); yvonne.simren@vgregion.se (Y.S.); hakan.bostrom@vgregion.se (H.B.); hanna.hebelka@vgregion.se (H.H.); 2Institution of Clinical Sciences, Sahlgrenska Academy, University of Gothenburg, 41345 Gothenburg, Sweden; 3Department of Pediatric Medicine, Sahlgrenska University Hospital, 41345 Gothenburg, Sweden; nils.ekvall@vgregion.se; 4Department of Clinical Pathology, Sahlgrenska University Hospital, 41345 Gothenburg, Sweden; maja.ostling@rmv.se (M.Ö.); lise-lotte.sten@vgregion.se (L.S.); 5Department of Medical Physics and Techniques, Sahlgrenska University Hospital, 41345 Gothenburg, Sweden; kerstin.lagerstrand@vgregion.se

**Keywords:** ultrasound, liver, biopsy, steatosis, inflammation, fibrosis, dispersion

## Abstract

Aim: To evaluate the feasibility of multiple ultrasound markers for the non-invasive characterization of fibrosis, inflammation and steatosis in the liver in pediatric patients. Materials and methods: The quantitative ultrasound measures shear wave elastography (SWE), shear wave dispersion (SWD) and attenuation imaging (ATI) were compared and correlated with percutaneous liver biopsies and corresponding measures in a control cohort. Results: The median age of the 32 patients was 12.1 years (range 0.1–17.9), and that of the 15 controls was 11.8 years (range: 2.6–16.6). Results: There was a significant difference in SWE values between histologic grades of fibrosis (*p* = 0.003), with a positive correlation according to the grade (r = 0.7; *p* < 0.0001). Overall, a difference in SWD values between grades of inflammation was found (*p* = 0.009) but with a lack of correlation (r = 0.1; *p* = 0.67). Comparing inflammation grades 0–1 (median:13.6 m/s kHz [min; max; 8.4; 17.5]) versus grades 2–3 (16.3 m/s kHz [14.6; 24.2]) showed significant differences between the groups (*p* = 0.003). In the 30 individuals with a steatosis score of 0, ATI was measured in 23 cases with a median value of 0.56 dB/cm/MHz. Conclusion: Comprehensive ultrasound analysis was feasible to apply in children and has the potential to reflect the various components of liver affection non-invasively. Larger studies are necessary to conclude to what extent these image-based markers can classify the grade of fibrosis, inflammation and steatosis.

## 1. Introduction

Chronic liver disease in pediatric cohorts is an important and increasing health issue associated with progressive fibrosis and cirrhosis [1]. With the development of new and individualized treatment strategies in both medical and surgical fields, the need for detailed characterization of liver tissue is increasing [1,2,3]. Currently, the reference standard for liver tissue characterization is liver biopsy [2,3,4]. However, biopsies are invasive procedures with multiple disadvantages, such as high cost, pain, risk of bleeding, sampling error, poor availability and, in children, the need for anesthesia [4]. 

Advances in ultrasound technology allow for the non-invasive and quantitative characterization of liver tissue. To estimate liver fibrosis, shear wave elasticity (SWE) is used, since decreased elasticity, i.e., a stiffer liver, correlates with an increased stage of fibrosis [2,3,5,6,7]. The recently available ultrasound measure shear wave dispersion (SWD) [8,9] uses multifrequency tones to assess dispersions of shear wave speed, which has been reported to be related to tissue viscosity and thus seems to reflect components of inflammation, edema and necrosis [9,10,11,12]. The few published studies, both animal and small clinical cohort studies in adult patients, report that SWD has an advantage over SWE in determining the degree of inflammation [6,8,13,14,15]. The attenuation imaging coefficient (ATI) is an additional tool enabling a surrogate estimation of fat content in the liver. Several adult studies have shown a good diagnostic capability to detect and stage liver steatosis using ATI [1,7,16,17,18].

Studies evaluating the combined use of SWE, SWD and ATI for the detailed characterization of the liver in pediatric patients are lacking. If characterization of the liver using these quantitative markers could be used to non-invasively distinguish the involvement of fibrosis, inflammation and steatosis in the liver, it would be an important clinical tool to diagnose and monitor liver disease. 

The purpose of this study was to evaluate the feasibility of comprehensive ultrasound analysis in pediatric patients for the characterization of fibrosis, inflammation and steatosis of the liver by comparing SWE, SWD and ATI with liver biopsies and a control cohort.

## 2. Materials and Methods

### 2.1. Liver Measurements

All patients (age 0–18 years) scheduled for a clinically indicated liver biopsy by the medical department at the Queens Silvia’s Children’s Hospital, Gothenburg, Sweden, between May 2021 and November 2021 were consecutively invited to participate in this prospective feasibility study. Inclusion criteria were any liver disease, suspected (in terms of increased serological markers to justify biopsy) or confirmed, and consent to participate in the study from the child/guardians. In all patients, measurements of liver elasticity (SWE), dispersion (SWD) and attenuation (ATI) were primarily performed during anesthesia (fasting > 4 h) and free breathing by one of five pediatric radiologists with elastography training [19]. When possible, i.e., with a cooperating patient, all liver measurements were also obtained in awake state, right before anesthesia. Two-dimensional SWE (Canon Medical, Aplio i800 (Tokyo, Japan)) measurements were performed in the right liver lobe with an intercostal approach, applying minimal transducer (curved transducer i8CX1) pressure. Since SWD (m/s/kHz) is obtained simultaneously with SWE (kPa) measurements, they were performed according to the latest liver elastography consensus statement by the Society of Radiologists in Ultrasound [9]. The patients were examined in a supine position with the right arm raised over their head. Whenever intercostal sampling was not possible, as in partial liver transplants, a subcostal acquisition was performed at a midline position. Liver measurements were performed 1.5–3 cm below the liver capsule using continuous mode, where the median of 10 registrations was recorded [9]. The measurements for obtaining ATI followed the same methodology, with a median of five registrations obtained according to recommendations. The quality estimate for ATI measurements has been reported to be excellent if R^2^ is ≥0.9 and good if ≥0.8 [20]. Since measurements are sometimes difficult to obtain in small pediatric livers, an R^2^ of ≥0.8 was deemed sufficient for inclusion. During the same anesthesia, following the SWE measurement, 1–2 percutaneous 18-gauge biopsies were obtained from the corresponding area of the liver. Exclusion criteria were median SWE values with IQR/median >30% kPa (also indirectly resulting in exclusion of corresponding SWD measurements) and median ATI R^2^ < 0.80.

During the same period, controls were consecutively included. To obtain similar conditions for the control cohort (anesthetized, fasting, etc.), children (0–18 years) without any known liver disease (or conditions/medications potentially affecting the liver) were recruited among patients planned for a clinically indicated biopsy of the kidney. The methodology for the controls followed the same principles as for the patients. 

The obtained measures (SWE/SWD/ATI) were compared to histology, hepatic serological markers, age and BMI. With few exceptions, all serological markers were collected on the same day as the biopsy or 1–2 days before. For analysis, liver measures obtained during anesthesia were primarily used; however, in the few cases where these did not fulfill the inclusion criteria, measures in the awake state were used. In addition, measures obtained during anesthesia and awake state were compared. The indications for liver biopsy/diagnosis of the patients are presented in Table 1. If the indication for clinical biopsy was only increased serological liver markers without an established liver disease, these patients were allocated to the unspecified group (Table 1). 

### 2.2. Histopathology Scoring

The pathologists were blinded to elastography results, and the physicians performing the liver measurements were blinded to histopathological results.

### 2.3. Grading of Fibrosis

Fibrosis severity was scored according to the Batts and Ludwig classification (stages 0–4 = F0–F4) [19], the scoring system clinically used at the study site to evaluate all liver biopsies. Blinded to the SWE measurements, one of two board-certified liver pathologists (7 and 4 years of experience, respectively) conducted the scoring of all biopsies. 

### 2.4. Grading of Inflammation

Inflammation was primarily categorized according to the Batts and Ludwig classification grades 0–4 in specimens with unspecified or chronic hepatitis [21] or according to the NAFLD score (0–3) in specimens with steatohepatitis. At the study site, an in-house descriptive scoring system is used clinically for unspecified inflammation accordingly: none (0), mild (1), moderate (2) and severe (3) [19]. To be able to compare the ultrasound measures with a single inflammation score, this in-house scoring was used both when unspecified inflammation was present in the specimen and for conversion from the Batts and Ludwig classification and the NAFLD score. The in-house descriptive scoring system corresponds to the Batts and Ludwig classification, that is, grade 0 (none), grade 1 (mild), grades 2–3 (moderate) and grade 4 (severe), and to the NAFLD score, that is, grade 0 (none), grade 1 (mild), grade 2 (moderate) and grade 3 (severe). 

### 2.5. Grading of Steatosis

The biopsies were graded according to the NAFLD score with estimation of percentage of liver cells affected: <5% (score 0), 5–33% (score 1), 34–66% (score 2) and >66% score 3.

### 2.6. Reliability Valuation

Inter-observer reliability regarding SWE measurements (also indirectly for SWD) for the same physicians has previously been reported with excellent results, as was intra-observer reliability measurements on previously obtained multi-mode cine-loops SWE sampling [19]. Reliability measures for ATI measurements were not performed as part of this feasibility study but have been reported to be high in adult cohorts [22].

### 2.7. Statistics

Descriptive statistics were used, with *n* (%) presented for categorical variables and median (min; max) presented for continuous variables. For pairwise comparison between groups of continuous variables, Fisher’s non-parametric permutation test or Mann–Whitney U was used. For comparison between groups of dichotomous variables, Fisher’s exact test was used. To test for association between SWE, SWD and ATI values and grade of fibrosis, grade of inflammation and steatosis, Spearman rank correlation coefficient (rho) was used.

Logistic regression analysis was performed for the variables SWE, SWD and ATI to predict the outcome. Area under ROC curve (AUC statistics) was calculated for description of goodness of predictors when applicable. The data were analyzed using version 9.4 of the SAS System. 

A *p*-value < 0.05 was considered significant in this analysis, and multiple testing corrections were not performed because this was an exploratory analysis.

## 3. Results

### 3.1. Patient Characteristics and Liver Measures

During the inclusion period, 35 patients were asked to participate, of whom two patients/guardians declined participation; hence, quantitative liver measures and biopsies were performed on the same occasion in 33 patients. Due to IQR/median > 0.30 kPa in both awake and anesthetized states, one patient was excluded. The median age for the 32 patients included was 12.1 years (range 0.1–17.9 years), with 69% males. 

The distribution of obtained SWE/SWD measures in anesthetized and awake states is displayed in Table 2. The liver measures SWE and SWD were obtained in both the awake state and during anesthesia in most cases (Table 2); however, in five cases, the measures in the awake state were used for analysis due to IQR/median > 0.30 kPa during anesthesia. Due to initial technical problems and human error with ATI sampling, ATI was not obtained at all in nine of the patients. Comparing quantitative liver measures sampled in awake and anesthetized states did not show any significant difference (Table 2).

Descriptive data for patients and controls, as well as the distribution of the grade of fibrosis, inflammation and steatosis within the patient cohort, are displayed in Table 3. 

### 3.2. Liver Biomarkers and Histopathology

#### 3.2.1. SWE

Overall, there was a significant difference found in SWE values between grades of fibrosis (*p* = 0.003). The difference was significant for all grades (0.03 > *p* < 0.002) except between F0 and F1 (*p* = 0.056) and between F2 and F3 (*p* = 0.50) (Figure 1). There was a significant positive correlation between the grade of fibrosis and SWE (r = 0.7; *p* < 0.0001), a positive weak correlation between the grade of fibrosis and SWD (r = 0.4; *p* = 0.02) and no correlation with ATI (r = −0.08; *p* = 0.73). The area under the receiver operating curve (AuROC) for differentiating F0–F1 from F2–F4 was 0.891 (95% CI: 0.708–1.0). A cut-off SWE value of ≤4.7 kPa yielded 100% sensitivity and 86% specificity to rule out F2–F4. 

#### 3.2.2. SWD

Overall, there was a significant difference found in SWD values between grades of inflammation (*p* = 0.0085), with higher values in grade 0 as compared with grade 1, while grade 2 and grade 3 displayed higher values compared to grades 0–1 (Table 3, Figure 1). No correlation between grades of inflammation and SWD (r = 0.1; *p* = 0.67) was found, nor was there any correlation with ATI (r = −0.1; *p* = 0.80), but there was a weak positive correlation with SWE (r = 0.4; *p* = 0.049). Comparing the dichotomized grade of inflammation, i.e., grades 0–1 (median 13.6 [8.4; 17.5] m/s/kHz) versus grades 2–3 (median 16.3 [14.6; 24.2] m/s/kHz) showed a significant difference between the groups (*p* = 0.0028), reflecting higher viscosity in the latter. 

#### 3.2.3. ATI

In 30 individuals with a steatosis score of 0, median ATI was measured in 23 cases, displaying a median ATI of 0.56 (0.4; 0.94) dB/cm/MHz and 0.82 dB/cm/MHz in one individual with a steatosis score of 1. One additional individual had a steatosis score of 1, but ATI was not measured. Correlation analysis between steatosis and ATI was omitted due to a lack of sufficient patients displaying any steatosis in their specimens.

### 3.3. Image-Based Liver Biomarkers and Serological Markers

Significant differences between patients and controls were found regarding thrombocytes, white blood cell count and ALT (Table 4). For all other serological markers, no differences between the groups were discovered. Four of the patients had cholestasis according to their biopsies. No correlation was found between the image-based liver biomarkers and serological markers, age or BMI.

### 3.4. Patients with Increased Serological Liver Markers without an Established Liver Disease

Eight patients underwent a biopsy for suspected liver disease. According to liver specimens, three had no signs of liver disease, one was shown to have iron deposition disease, and the remaining four had contributing factors that were interpreted as the cause of increased serological liver markers, such as end-stage renal failure causing death, ulcerous colitis, drug side effect and worm infection.

### 3.5. Controls

Fifteen controls were invited, all of whom accepted participation. The median age was 10.6 years (range: 2.6–16.6 years), with 58% males (Table 4). The measures SWE/SWD were obtained in both the awake state and during anesthesia in 10 of the 15 controls. All measures during anesthesia were used except for in one individual, where the measures in the awake state were included in the analysis. The reason for excluding the measures in the sedated state for this individual was IQR/median > 0.30 kPa. No individual was entirely excluded. In four of the controls, ATI was not obtained at all. ATI was obtained in both states in six individuals, only during anesthesia in three and only in the awake state in two (Table 4). The most common indications for clinical biopsy in the control group were investigation of nephrosis (30% of cases), followed by nephritis/hematuria (20%) and transplant rejection (20%), and, in addition, a small number were due to unclear kidney failure and follow-up due to glomerulonephritis. A clinician (N.E.) carefully controlled all clinical and drug information about the patients in the control group to rule out any suspicion of liver disease or medications potentially affecting the liver.

### 3.6. Comparison between Patients and Controls

No significant difference in sex, age, weight, height, BMI or ATI was found between the groups (Table 4). Significantly higher SWE and SWD values were found in patients compared to controls (Table 4), reflecting increased elasticity and viscosity in the patient group. There was no significant difference in median ATI between the cohorts. Significant differences between the groups existed in the serological markers ALT, white cell count and thrombocytes (Table 4).

Patients without fibrosis (F0) had a median SWE value of 4.9 (2.9; 8.1) kPa, which did not differ compared to the control group, which displayed a median SWE value of 4.6 (3.3; 7.5) kPa (*p* = 0.67). Patients with inflammation grade 0 had a median SWD of 14.6 (10.1; 17.5) m/s/kHz, which was significantly higher compared to the median of 11.7 (9.4; 13.7) m/s/kHz in controls (*p* = 0.0008). Comparing patients with steatosis grade 0 (median 0.56 [0.4; 0.94] dB/cm/MHz) with controls (median 0.54 [0.45; 0.85] dB/cm/MHz) displayed no significant difference (*p* = 0.87) (Table 4).

## 4. Discussion

In this prospective study, the feasibility of using multiple ultrasound-based markers to reflect liver affections of fibrosis (SWE), inflammation (SWD) and steatosis (ATI) with histologic correlation was, to our knowledge, explored for the first time in a pediatric cohort with liver disease. A strong significant correlation was observed between the grade of fibrosis and SWE and, to a lesser extent, SWD. Patients with moderate/severe inflammation (grades 2–3) displayed significantly higher SWD values as compared to patients with no or mild inflammation (grade 0–1), reflecting higher liver viscosity in the former group. No further significant correlation between SWD and inflammation could be established.

Most human organs change during pathological processes and thereby change their viscoelastic properties. It is therefore appealing to use non-invasive biomarkers that reflect both the tissue elasticity as well as the tissue viscosity, since these reflect different properties of the tissue [14,20]. The ability of SWE to assess liver stiffness, a reflection of liver fibrosis, is an established method, at least in adults, while more studies in the pediatric population are still warranted before it can be fully used in clinical routine [19]. Most ultrasound models to assess tissue elasticity are linear models; however, in tissues with dispersive properties, the speed and attenuation of the shear wave increase with frequency [8,9,23]. Analysis of the dispersion slope of the shear waves, generating the SWD value, has been reported in a few preclinical and clinical studies as a method to reflect tissue viscosity [14,20,23,24,25]. For example, in a study of rat livers, Sugimoto et al. showed that SWE was a more effective predictor of fibrosis than SWD, while SWD was a more effective predictor of the grade of necroinflammation in comparison to SWE [14], results at least partly supported by this feasibility study in pediatric patients. The scarcity of published data on SWD makes the comparison and interpretation of the measured values difficult. To date, SWD has been studied in a handful of adult cohorts [7,13,23,24,25,26,27]. Only three of these related SWD to histology, and since only specific cohorts were studied, i.e., liver transplant recipients and NAFLD patients, the results cannot be generalized, but results indicate, similarly to the current study, that SWD seems to be useful for assessing inflammation in liver disease [24,25,28]. 

To our knowledge, only Trout et al. have investigated liver SWD in a pediatric cohort. They studied SWE and SWD in 128 healthy children, though without histologic correlation, and reported promising results [7]. In alignment with our control cohort, with a mean SWD of 11.9 m/s/kHz, Trout et al., using the same ultrasound machine as in the current study, reported a mean SWD of 11.4 m/s/kHz in their healthy cohort [7]. In our cohort, patients without inflammation displayed a mean SWD of 14.5 m/s/kHz. This finding suggests that inflammation is not the only determinant for increased viscosity, since other factors such as edema, cholestasis and necrosis also affect the dispersion slope [9,29,30]. Similarly, Lee et al. reported, in adult liver transplant recipients, that both the grade of fibrosis and inflammatory activity were determinant factors for SWD. Lee et al. concluded that SWD provided better diagnostic performance of allograft damage as compared to the SWE value alone [13], likely because the involvement of inflammation is also reflected. In alignment with Schultz et al., no correlation between SWD and serological biomarkers was found [26]. Even if other factors seemingly affect the SWD value, this marker seems promising for appreciating the involvement of liver inflammation. 

Furthermore, viscosity in the pediatric liver seems to be higher in general as compared to adults. Sugimoto et al. proposed, in their biopsy-controlled adult cohort, an SWD cutoff > 9.9 m/s/kHz for grade 2 inflammation and >12.5 m/s/kHz for grade 3 inflammation [25]. The median SWD in our entire biopsy-controlled pediatric patient cohort was 14.4 (8.4; 24.2) m/s/kHz, although only six patients had higher than grade 1 inflammation. One likely reason for this discrepancy is different age-dependent viscoelastic properties. Liver stiffness values are lower in children than in adults, with greater differences at higher frequencies. As part of normal maturation, the stiffness increases with age, reaching the adult level in late adolescence, with, for example, increased collagen content in the adult liver as compared to in children [29,31]. Therefore, it seems reasonable that not only the elasticity but also the viscosity differs between adults and children. It is likely necessary to stratify for age to adequately interpret SWD values.

The current finding of the median SWE value of 4.9 kPa for the pediatric liver without fibrosis confirms a recent published pediatric study using the same ultrasound machine, with correlation with same-day histology suggesting a cut-off of median ≤ 4.5 kPa to rule out significant fibrosis [19]. Applying a cut-off ≤ 4.7 kPa in the current study provided 100% sensitivity and 86% specificity to rule out significant fibrosis (F0–F1) with a high diagnostic performance. An example of one of our patients with a biopsy-verified healthy liver is displayed in Figure 2A. The results are also in alignment with the study in healthy children by Trout et al., with a mean SWE of 1.29 m/s, corresponding to a mean of approximately 5 kPa [7]. The finding that the SWE value is affected by both fibrosis and inflammation is well known. For example, various forms of hepatitis alter the SWE value despite being the same grade of fibrosis [20,30]. 

Considering that 43% of our patient cohort did not have any form of inflammation in their livers, and 41% of the patients only had mild inflammation, the impact on SWE with coexisting fibrosis and inflammation is uncertain. This needs to be addressed in larger studies.

In our cohort, only four individuals had signs of cholestasis in their specimens. Interestingly, all these patients had high SWD values, and all, except one who displayed severe fibrosis, displayed grade 2 inflammation on biopsy. One of these was a patient with suspected autoimmune hepatitis with increased serological hepatic markers (AST/ALT) in whom conventional ultrasound, however, displayed slightly marked intrahepatic bile ducts, and a diagnosis of primary sclerosing cholangitis was made (Figure 2B). SWD, thus, seems to reflect the involvement of both cholestasis and inflammation. However, future studies need to elucidate to what extent the respective components affect SWD. 

The ATI technique has been shown to be promising for the non-invasive diagnosis and quantification of hepatic steatosis in adults [18,31,32]. Bea et al. suggested a cut-off for no steatosis of <0.63 dB/cm/MHz in their biopsy-controlled cohort of over 100 adults [28], a cut-off supported by others [18,33]. As with SWD, pediatric studies on ATI are very scarce, and to our knowledge, no studies in children with disease exist, nor have any biopsy-controlled studies been published. Recently, Cailloce et al. investigated 86 children without known liver disease and suggested a median ATI coefficient of 0.65 dB/cm/MHz as a cut-off for a healthy liver, with the reservation that no biopsy reference existed. The ATI values in children without steatosis seem to be slightly lower as compared to the adult population. Our control group had a median of 0.54 (0.45; 0.85). Eliminating the one patient with steatosis, the biopsy-controlled ATI for steatosis grade 0 was a median of 0.56 (0.4; 0.94), which can serve as a cut-off in the pediatric population for when steatosis is very unlikely. The lack of steatosis in the current cohort precludes a conclusion regarding the diagnostic performance of ATI to predict steatosis. One patient in the current study was referred for biopsy due to increased serological markers and high BMI (29). The ATI was high (0.82 dB/cm/MHz), and SWE was slightly increased (6.60 kPa), as was SWD (11.7 m/s/kHz), with biopsies revealing a steatosis score of 1, fibrosis grade 1 and inflammation grade 1.

Today, clinical management of many liver diseases requires a biopsy to assess histological features such as fibrosis, inflammation and steatosis. However, especially in children who require anesthesia for biopsy, the drawbacks are many, which is why non-invasive and easily applied methods to assess different tissue characteristics are desirable. The use of multiple ultrasound-based markers (SWE/SWD/ATI) provides additional pathophysiological information, as compared to individual measures or conventional ultrasound alone. The method shows great potential, on a group level, to aid in estimating steatosis, inflammation and fibrosis of the liver non-invasively in one single examination, but its capacity to predict the various components of disease on an individual level needs to be investigated. 

## 5. Limitations

With the inherent limitations of a feasibility study with a small sample size, stratification for different grades of fibrosis/inflammation/steatosis was not possible, nor was stratification for age or disease. However, it can also be considered a strength that the overall feasibility of the method to assess pediatric patients with liver disease, in whom disease is not always known beforehand, was investigated. The absence of scientifically designed guidelines on how to ensure proper SWD measurements is of course a limitation. The risk of sampling error does exist, which, however, is also true for biopsies. Reliability measures for ATI measurements were not performed as part of this feasibility study but have been reported to be high in adult cohorts [22]. The lack of consistency in the sampling of measures in the awake or sedated state is a limitation, since SWE has been reported in some studies to vary between awake and sedated states [33]. Therefore, we tried to obtain measures in both states when possible, and the analysis did not reveal any significant differences between the states on a group level, even if differences sometimes existed on an individual level. 

## 6. Conclusions

Comprehensive ultrasound analysis using the quantitative markers SWE, SWD and ATI was feasible in children, including during free breathing, and has the potential to reflect the various components of liver affection non-invasively. These image-based markers could likely be used clinically to rule out significant fibrosis, inflammation and steatosis in children. However, this needs to be established in larger studies.

## Figures and Tables

**Figure 1 children-09-00692-f001:**
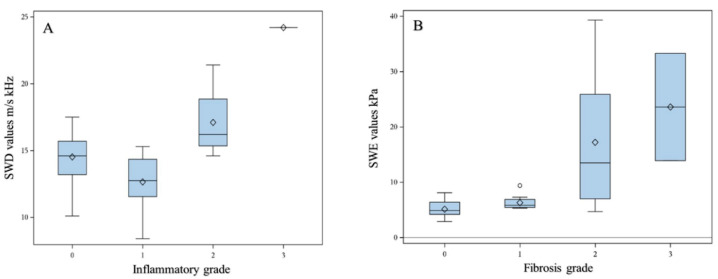
Box and whisker plot diagram for SWD (**A**) and SWE (**B**) relative to inflammation grade and fibrosis grade.

**Figure 2 children-09-00692-f002:**
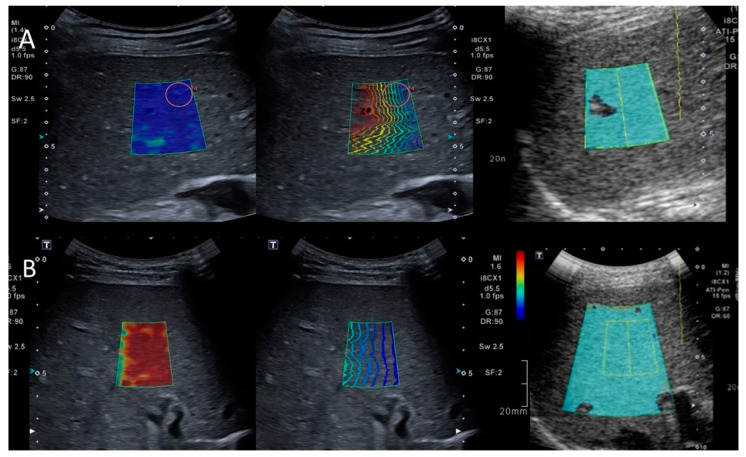
(**A**): An 11-year-old child with ulcerous colitis and autoimmune hepatitis under treatment. The colormap and SWE (left/middle) display homogeneous color and normal propagation of the shear waves with all markers low: SWE 3.9 kPa, SWD 10.5 m/s/kHz and ATI 0.57 0.53 db/cm/MHz. The biopsy showed no signs of fibrosis, inflammation or steatosis. (**B**): Previously healthy 15-year-old child with increased AST/ALT. SWE (left) displays intense red, inhomogeneous color with increased distance between the shear waves (middle). Both SWE (39 kPa) and SWD (middle) 21 (m/s/kHz) values were high. ATI (right) was low (0.53 db/cm/MHz). Biopsy revealed grade 2 fibrosis and inflammation, but no steatosis.

**Table 1 children-09-00692-t001:** Indications for Biopsy/Clinical Diagnosis.

	Number of Patients
Liver transplants, yearly check-ups and acute indications	8
Unspecified—increased serological liver markers	8
Autoimmune hepatitis	7
Alfa-1-antitrypsin deficiency	4
Cholestatic disease	2
Primary and autoimmune sclerosing cholangitis	1
Non-alcoholic steatohepatitis (NASH)	1
Steatosis	1

**Table 2 children-09-00692-t002:** Distribution of obtained ultrasound-based markers in anesthetized and awake patients.

	Under Anesthesia	Awake	*p*-Value
SWE (kPa)	5.4 (2.9; 39.3) *n* = 41	5.4 (3; 20.1) *n* = 25	0.17
SWD (m/s kHz)	13 (8.4; 24.2) *n* = 41	13.6 (9.4; 16.9) *n* = 25	0.15
ATI (dB/cm/MHz)	0.56 (0.45; 0.94) *n* = 27	0.54 (0.38; 0.85) *n* = 19	0.14

Values in median (min; max), *n* (%) for categorical variables; SWE = shear wave elastography; SWD = shear wave dispersion; ATI = attenuation imaging.

**Table 3 children-09-00692-t003:** Quantitative ultrasound markers stratified according to histological grading in patients.

**Fibrosis Grade**	***n* (%)**	**SWE (kPa) Median (Min; Max)**
0	10 (31.3)	4.9 (2.9; 8.1)
1	12 (37.5)	5.9 (5.3; 9.4)
2	7 (21.9)	13.5 (4.7; 39.3)
3	3 (9.4)	23.6 (13.9; 33.3)
**Inflammation grade**		**SWD (m/s/kHz)**
0	14 (43.8)	13.1 (8.4; 17.2)
1	13 (40.6)	13.6 (11.3; 17.2)
2	4 (12.5)	16.1 (10.7; 24.2)
3	1 (3.1)	15.8 (15.3; 16.3)
**Steatosis score**		**ATI (dB/cm/MHz)**
0	22 (95.6)	0.56 (0.4; 0.94)
1	1 (4.4)	0.82

SWE = shear wave elastography; SWD = shear wave dispersion; ATI = attenuation imaging.

**Table 4 children-09-00692-t004:** Demographic data and serological and ultrasound-based markers in patients and controls.

	Patient (*n* = 32)	Control (*n* = 15)	*p*-Value
Male	22 (68.8%)	8 (53.3%)	0.48
Female	10 (31.3%)	7 (46.7%)	
Age (years)	12.1 (0.1; 17.9)	11.8 (0.1; 17.9)	0.75
Height (cm)	148.5 (57; 191.3)	149 (86; 177)	0.78
Weight (kg)	37.5 (4.5; 96)	44 (11; 71)	0.91
BMI (kg/m^2^)	17.7 (13.5; 28.9)	17.7 (13; 26.7)	0.78
INR (prothrombin time)	1.05 (0.9; 1.6)	1 (0.9; 1.2)	0.12
AST (µkat/L)	15 (3; 15)		N/A
ALT (µkat/L)	0.8 (0.22; 16)	0.14 (0.1; 0.4)	<0.005
White cell count (×10^9^/L)	5.4 (2.2; 12.2)	8.4 (5.1; 25.5)	<0.035
Thrombocytes (×10^9^/L)	239 (44; 444)	333 (149; 707)	<0.02
Gamma-GT (µkat/L)	0.53 (0.15; 10)	0.83 (0.16; 1.5)	0.9
Bilirubin (µkat/L)	8.6 (3.3; 357)	6.2 (5; 7.2)	<0.17
SWE (kPa)	6.2 (2.9; 39.3)	4.6 (3.3; 7.5)	<0.002
SWD (m/s/kHz)	14.4 (8.4; 24.2)	11.7 (9.4; 13.7)	<0.005
ATI (dB/cm/MHz)	0.56 (0.4; 0.94) *n* = 23	0.54 (0.45; 0.85) *n* = 11	0.87

Values in median (min; max), *n* (%) for categorical variables; INR = international normalized ratio; AST = aspartate transaminase; ALT = alanine aminotransferase; BMI = body mass weight. SWE = shear wave elastography; SWD = shear wave dispersion; ATI = attenuation imaging.

## Data Availability

The datasets used and/or analysed during the current study are available from the corresponding author on reasonable request.

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
