# Peer review of "Ultrasound Shear Wave Elastography, Shear Wave Dispersion and Attenuation Imaging of Pediatric Liver Disease with Histological Correlation"

_children, 2022, doi:10.3390/children9050692_

Round 1

Reviewer 1 Report

This interesting study that included 35 pediatric patients aimed to evaluate the feasibility of multiple ultrasound markers for the characterization of fibrosis, inflammation, and steatosis in the liver non-invasively. Liver biopsy still is currently the gold standard method for evaluating liver fibrosis, steatosis, and inflammation. But, recently new software embedded in ultrasound devices have been developed. Particularly, non-invasive multiparametric ultrasound-based tools that can quantify steatosis and fibrosis are already used in clinical practice.  The hepatic shear wave speed would increase uniformly with increasing fibrosis in a simple and precise manner, not influenced by other factors. However, biological soft tissues are rather viscoelastic, than entirely elastic. Therefore, the role of cofactors can be major. In addition to liver fibrosis, several factors, such as the presence of steatosis or necro-inflammation influence the viscoelastic properties of the liver tissue. Given these pathological conditions often coexist it is very important to better study, analyze and understand these newly developed parameters. As the authors pointed out in the limitations section, the small number of patients included in this study is the most important drawback. But going beyond the obvious limitations of this study, I consider that the results were well presented and are a plus to the current knowledge in this field.

I suggest adding this study that evaluates the combined use of SWE, viscosity, and attenuation for the characterization of  liver pathology in adult individuals to the reference list:

 Popa, A., Bende, F., Șirli, R., Popescu, A., Bâldea, V., LupuÈ™oru, R., Cotrău, R., Fofiu, R., Foncea, C., & Sporea, I. (2021). Quantification of Liver Fibrosis, Steatosis, and Viscosity Using Multiparametric Ultrasound in Patients with Non-Alcoholic Liver Disease: A "Real-Life" Cohort Study. Diagnostics (Basel, Switzerland), 11(5), 783. https://doi.org/10.3390/diagnostics11050783

Author Response

Dear reviewer, thank you for your wise remarks to our manuscript.

Response 1:

We have added the abovementioned study in the introduction (reference 9). Additionally we found a book chapter discussing elastography interesting, titled "Ultrasound Based Elastography Techniques for the Evaluation of Nonalcoholic Liver Disease” by Sporea, Ioan, Raluca LupuÈ™oru, and Roxana Șirli, 2022, and added it as reference number 12.

Reviewer 2 Report

This is a well written and interesting paper, although small in sample size

Introduction-Good summary and explanation of the topic and the modalities used to asses these areas

Methods- 

It is good that this is a prospective study, the methods are clear and well described.

2 comments:

The recruitment period could have been longer and this should be recognised in the discussion.

It would have been more definitive to include only those with confirmed liver disease, 'suspected' liver disease may be at any stage in the diagnosis process, or had the chance to develop chronicitiy.

Further information should be given on the 8 in the 'unspecified' group- was biopsy done because of raised transaminases- what was found? 'non specific hepatitis'

Results:

31% of patients and 43% of patients had no inflammation, this removes a reasonable number of patients who could have their measurements compared to assess for fibrosis and inflammation and hence a larger sample size may have given more information.

Otherwise clearly written results

Discussion:

Please mention the above points as part of limitations, otherwise clearly thought out discussion

Author Response

Response to Reviewer 2 Comments

Dear reviewer, thank you for your wise remarks to our manuscript. We have made minor revisions of the manuscript after your suggestions, please see below:

Point 1: The recruitment period could have been longer and this should be recognised in the discussion.

Response 1:

We agree that the inclusion period would have benefited from being extended, as well as that the sample size would have been larger. However, we wanted to evaluate feasibility in pediatric patients before continuing with the planned large-scale study and believe that the results are interesting enough to be disseminated further. Upon your suggestion we have rewritten and clarified this in the limitations section of the study, please see our new text below:

Limitations

With the inherent limitations being a feasibility study with a small sample size, stratification for different grades of fibrosis/inflammation/steatosis was not possible, neither stratification for age nor disease. However, it can also be considered a strength that the overall feasibility of the method to approach pediatric patients with liver disease, in whom disease not always is known forehand, was investigated. Absence of scientifically designed guidelines how to ensure proper SWD measurements is of course a limitation. Risk for sampling error do exist, which however is also true for biopsies. Reliability measures for ATI measurements were not performed as part of this feasibility study but has been reported high in adult cohorts [22]. The lack of consistency in sampling of measures in awake or sedated state is a limitation, since SWE has been reported in some studies to vary between awake and sedated state [37]. Therefore, we tried to obtain measures in both states when possible and analysis did not reveal any significant dif-ference between the states on a group level even if differences sometimes existed on an individual level. 

Point 2:

It would have been more definitive to include only those with confirmed liver disease, 'suspected' liver disease may be at any stage in the diagnosis process, or had the chance to develop chronicitiy.

Further information should be given on the 8 in the 'unspecified' group- was biopsy done because of raised transaminases- what was found? 'non specific hepatitis'

Response 2:

According to the study design, everyone who has a sufficiently suspected disease to justify a biopsy was included. The reason for this methodology was because we wanted to find out if, in the future, unnecessary biopsies potentially could be avoided by applying these parameters.

All in the "unspecified" group had elevated transaminases, of which biopsy in 2 of the cases showed absence of fibrosis, inflammation and steatosis in the liver specimen. The remaining 6 had various underlying causes for suspicions of liver disease. We have clarified and detailed this part of the article by adding a new subsection to Results titled “Patients with increased serological liver markers without an established liver diseaseto the manuscript, please see text below:

“3.4 Patients with increased serological liver markers without an established liver disease

Eight patients underwent a biopsy for suspected liver disease. Acording to liver specimen three had no signs of liver disease, one was shown to have an iron deposition disease and remaining four had contributing factors that were interpreted as the cause of increased serological liver markers such as end stage renal failure causing death, ulcerous colitis, drug side effect and a worm infection.“

Point 3:

31% of patients and 43% of patients had no inflammation, this removes a reasonable number of patients who could have their measurements compared to assess for fibrosis and inflammation and hence a larger sample size may have given more information.

Response 3:

We have clarified this section in the discussion section by adding the following paragraph to the text:

“Considering that 43% of our patient cohort did not have any form of inflammation in their liver, and 41% of the patients only had mild inflammation, the impact on SWE with coexisting fibrosis and inflammation is uncertain. This needs to be adressed in larger studies.”